# A Lightweight Model for Ship Detection and Recognition in Complex-Scene SAR Images

**Boli Xiong, Zhongzhen Sun * , Jin Wang, Xiangguang Leng and Kefeng Ji**

College of Electronic Science and Technology, National University of Defense Technology, Changsha 410073, China
* Correspondence: sunzhongzhen14@nudt.edu.cn; Tel.: +86-178-7313-4710

**Abstract:** SAR ship detection and recognition are important components of the application of SAR data interpretation, allowing for the continuous, reliable, and efficient monitoring of maritime ship targets, in view of the present situation of SAR interpretation applications. On the one hand, because of the lack of high-quality datasets, most existing research on SAR ships is focused on target detection. Additionally, there have been few studies on integrated ship detection and recognition in complex SAR images. On the other hand, the development of deep learning technology promotes research on the SAR image intelligent interpretation algorithm to some extent. However, most existing algorithms only focus on target recognition performance and ignore the model's size and computational efficiency. Aiming to solve the above problems, a lightweight model for ship detection and recognition in complex-scene SAR images is proposed in this paper. Firstly, in order to comprehensively improve the detection performance and deployment capability, this paper applies the YOLOv5-n lightweight model as the baseline algorithm. Secondly, we redesign and optimize the pyramid pooling structure to effectively enhance the target feature extraction efficiency and improve the algorithm's operation speed. Meanwhile, to suppress the influence of complex background interference and ships' distribution, we integrate different attention mechanism into the target feature extraction layer. In addition, to improve the detection and recognition performance of densely parallel ships, we optimize the structure of the model's prediction layer by adding an angular classification module. Finally, we conducted extensive experiments on the newly released complex-scene SAR image ship detection and recognition dataset, named the SRSDDv1.0 dataset. The experimental results show that the minimum size of the model proposed in this paper is only 1.92 M parameters and 4.52 MB of model memory, which can achieve an excellent F1-Score performance of 61.26 and an FPS performance of 68.02 on the SRSDDv1.0 dataset.

**Keywords:** ship detection; ship recognition; lightweight model; attention mechanism; synthetic aperture radar (SAR)

## 1. Introduction

Radar is a kind of sensor that uses microwaves for active sensing. Synthetic Aperture Radar (SAR) extends the original concept of radar, using pulse compression technology and the principle of synthetic aperture, which enables the radar system to perform the two-dimensional high-resolution imaging of the target. Compared with optical and infrared sensors, SAR is not affected by weather, light, and other conditions. Therefore, SAR has the characteristics of all-day, all-weather, large-width, and high-resolution imaging, which has become a significant tool for Earth observation [1–3].

With the continuous development of SAR image resolution and image quality, the research on the automatic detection and identification of ship targets in SAR images has become a significant research direction in the domain of SAR image interpretation [4]. Among them, the SAR ship identification technology has involved the courtesies of a large number of researchers due to its task characteristics. In contrast, the research of ship

recognition technology started relatively late, and there have been little research results. In recent years, deep learning technology has accomplished encouraging results in various fields, including target detection [5–7], image classification [8,9], autonomous driving [10], saliency detection [11], semantic understanding [12], and so on [13–16]. Additionally, this new technique also provides new ideas for the development of SAR target detection and recognition technology. Additionally, it is possible to detect and recognize ship targets in SAR images automatically based on deep learning technology [17–20]. For example, Zhao et al. [21] proposed an automatic identification method for SAR ships based on feature decomposition across different satellites. This method improves the performance of target location and recognition by optimizing the backbone network to extract features. Yoshida et al. [22] proposed a method to automatically detect ships in motion based on the "You Only Look Once (YOLO) v5 model", which can effectively capture ship targets in ALOS-2 spotlight images. Zheng et al. [23] proposed an ensemble automated method (MetaBoost) for heterogeneous DCNN models based on two-stage filtering, which effectively achieved robustness and high-accuracy recognition for SAR ships. Extensive experiments on OpenSARShip and FuSARShip datasets show that the MetaBoost can significantly outperform individual classifiers and traditional ship recognition models.

As we all know, target detection and recognition have been developed and improved in natural scenes. However, SAR images have major differences from those in natural scenes, one of the most obvious points being that the change in the ship target's morphology and background interference, such as sidelobe, tailing, and so on. Although there is significant research on SAR image target detection, target classification, and recognition, they still have the following obvious problems:

- Due to the limitation of data quality and target labeling requirements, most of existing research methods are based on ship detection datasets such as SSDD [24], AirSARship [25], HRSID [26], and LS-SSDD-v1.0 [27]. Research on the integration of SAR ship detection and identification is lacking.
- The scenes of the SAR image are complex and changeable, which have obvious influence on target imaging and morphological changes. A large number of ships are missed and false alarms can easily occur nearshore or in areas where targets are densely distributed.
- Compared with traditional recognition models, most existing deep learning models show strong robustness and adaptability in target recognition performance. However, these methods have obvious shortcomings such as low model training efficiency, high deployment cost of embedded devices, and low real-time performance.

In response to the above problems, we propose a lightweight model for ship detection and recognition in complex-scene SAR images. The work of this paper mainly includes the following aspects:

1.  In order to improve the model's operating efficiency and reduce the cost of algorithm deployment, this paper improves and optimizes the algorithm based on the YOLOv5-n lightweight model. Combined with the fast pyramidal pooling structure, the target feature extraction efficiency of the neural network model is effectively improved.
2.  Aiming to improve the detection and recognition performance of ship targets in high-resolution complex-scene SAR images, this paper integrates an attention mechanism into the target feature extraction layer. The proposed attention module can improve the model's attention to target features in complex scenes and suppress the influence of background noise.
3.  To optimize the performance of ship positioning and recognition in complex scenes such as nearshore or the dense distribution of ship targets, this paper introduces an angle classification module in the prediction layer of the network model to achieve the rotation detection and recognition of ship targets.
4.  We conducted extensive experiments on the newly released SAR ship detection and recognition dataset named SRSDD [28] to validate the proposed improvements. The experimental results show that the proposed method in this paper not only

outperforms several other deep learning methods in terms of detection and recognition performance, but also has significant advantages in terms of algorithm parameters, model size, and operation efficiency.

The rest of this paper is organized as follows. In Section 2, the related research work is briefly introduced. In Section 3, we propose the overall framework of the model in this paper and describe several optimization modules. We analyze the experimental performance of the proposed lightweight model in Section 4. Meanwhile, some visual recognition results are presented in Section 4. Finally, the conclusions are drawn in Section 5.

## 2. Related Work

This section mainly introduces some of the latest deep learning methods applied in the field of SAR image ship detection and recognition. According to the improvement and innovation proposed by the methods in this study, the related methods are divided into three categories: lightweight models, embedded attention mechanism models, and rotation detection models.

### 2.1. Lightweight Models

The majority of convolutional neural network-based ship target detection algorithms improve the identification score at the expense of technical complexity, but they are challenging to directly apply and deploy. To address this problem, Zhou et al. [29] proposed a lightweight anchorless ship detection network (LASDNet) for SAR images. The method is improved based on the single-stage anchor-free network FCOS and can achieve the lowest parameter size of 1.15 MB, the lowest computational complexity (1.01 GFLOPs), and average precision (59.25) on the HRSID dataset. A lightweight framework based on a threshold neural network (TNN) was suggested by Cui et al. [30] to quickly locate ship targets in large-scene SAR imagery. The neural network, which has a higher detection accuracy and more resilience, is used in the procedure to obtain the target detection threshold. It is important to note that the techniques' size is considerably less than that of other deep learning models. An improved lightweight RetinaNet was recommended by Miao et al. [31] for ship detection in SAR images. The ghost module, which replaces the backbone's shallow convolutional layers and reduces the number of deep convolutional layers, is the network's main selling point because it allows for a significant reduction in the number of floating-point operations and parameters without sacrificing detection precision and recall. In their proposed learned detector, Light-YOLOv4, Ma et al. [32] eliminated the network's uninteresting channels and layers, considerably lowering the network's breadth and depth. OSCAR-RT, the first end-to-end algorithm/hardware codesign framework for SAR ship identification based on CNN on real-time satellites, was suggested by Yang et al. [33]. The model may concurrently produce highly effective field-programmable gate array (FPGA)-based hardware accelerators that can be installed on satellites, as well as precise, hardware CNN models. Yang et al. [34] proposed an efficient and lightweight object detection network combined with soft quantization. The network employs a soft quantization algorithm to reduce the impact of quantization errors on model accuracy. Yu et al. [35] suggested a fast and lightweight detection network named FASC-Net, and the model contains four kinds of network lightweight optimization modules, which greatly reduce the number of parameters of the target feature extraction network. Chang et al. [36] develop a new architecture with less layers called YOLOv2-reduced to reduce the computational time. Yu et al. [37] suggested a light ship identification network based on YOLOX-s. The network outputs a pyramid structure with a large amount of computation and builds a streamlined network on the first-level features, which effectively improves the detection efficiency. Feng et al. [38] proposed a new lightweight position-enhanced anchorless SAR ship detection algorithm LPEDet. The network uses YOLOX as the benchmark framework and redesigned a lightweight multi-scale backbone to balance the detection speed and accuracy. Li et al. [39] proposed an ultralight and high detection accuracy SAR ship detection method based on YOLOX. The network contains an ultralight and high-performance

detection backbone based on ghost cross stage partial (GhostCSP) and lightweight spatially dilated convolutional pyramid (LSDP). Xu et al. [40] designed a lightweight cross-level local (L-CSP) module to decrease the computation and prune the network for a more compact detector. Guo et al. [41] proposed a lightweight single-stage SAR ship target detection model, called Yold-based Lightweight multi-scale ship detector (LMSD-YOLO), in which the DSASFF module is designed, to attain the adaptive fusion of multi-scale features with a few parameters. Liu et al. [42] proposed a lightweight network based on the YOLOv4-LITE model, which uses MobileNetv2 to extract the features and designed an improved receptive field block (RFB) structure.

### 2.2. Embedding Attention Mechanism Models

Ship target recognition in SAR images frequently creates more false alarms and may result in the missing detection of tiny objects because of the complex backdrop and coherent speckle noise interference. Gao et al. [43] proposed an improved attention-based yolo4 (imyolo4) model. The adverse effects of the complex background and noise are suppressed by introducing a threshold attention module (TAM), and an attention module is added to the backbone network to enhance the discriminative ability of the multi-scale target features. Peng et al. [44] suggested a new approach based on enhanced YOLOX as an anchor-free target detection method, which achieved high-speed and high-precision ship target detection by combining a coordinated attention mechanism and improving the loss function. Zha et al. [45] proposed a novel ship detection model based on multi-feature transformation and fusion (MFTF-Net) to suppress false alarms and improve small target detection performance. To reduce the interference of noisy information, the method applies a modified convolutional block attention module (CBAM) and a squeeze-excited attention (SE) mechanism to the lower and upper two layers of the network output, respectively. Jiang et al. [46] proposed an effective lightweight anchor-free detector called R-Centernet+ and introduced the CBAM module to the backbone network. Zhang et al. [47] proposed a cross-scale region predictive perception network (CSRP-Net), by designing a cross-scale self-attention (CSSA) module, which suppressed the influence of noise and complex background and enhanced the detection ability of multi-scale targets. A multi-scale detection network for ships in SAR pictures was suggested by Zhang et al. [48] and is based on attention and weighted fusion. This technique, which is based on the YOLOv5 architecture, introduces a coordinate attention block to sharpen the position characteristics of ship targets and muffle the interference from complicated backdrops. Guo et al. [49] added the CBAM module and the BiFPN module to the YOLOv5 network, which enables it to fully learn the feature information of the spatial and channel dimensions, and enhance the information fusion transfer between multi-scale targets, and the problem of missed detection of multi-scale targets is well resolved. A novel anchor-free SAR target recognition technique, AFSar, was put forth by Wan et al. [50] based on multi-scale improved representation learning. This approach emphasizes the distinct strong scattering properties of SAR targets by combining channel and spatial attenuation processes. Li et al. [51] proposed a new deep learning network, the attention-guided balanced feature pyramid network (A-BFPN), and designed a channel attention-guided fusion network (CAFN) model to obtain the optimized multi-scale features, reducing severe aliasing effects in the mixed feature maps. Zhou et al. [52] suggested a multi-scale ship detection network (MSSDNet) method based on a small model size YOLOv5 (YOLOv5s) by introducing an improved backbone network Res2Net (MRes2) with a coordinate attention module (CAM)) for the multi-scale feature extraction in scale dimension to better represent the features. Su et al. [53] suggested a new spatial information integration network (SII-Net) detection method, and designed a channel location attention mechanism (CLAM) module to extract location information along two spatial directions to enhance the detection capability of the backbone network.

### 2.3. Rotation Detection Models

Ship targets are often densely arranged and docked in nearshore areas. The traditional positive frame detection method will cause a large number of missed detections and false detections of targets. In order to improve the detection performance of ship targets in the berth area in SAR images, Zhao et al. [54] proposed a ship rotation detection method based on an orientation-aware feature fusion network (OFF-Net) in SAR images. This method proposes a decoupled orientation perception head that is more robust to ship targets with arbitrary orientations. At the same time, the authors also provide a high-resolution SAR ship detection dataset (OBB-HRSDD) with rotatable bounding boxes. Man et al. [55] proposed a new center-to-corner vector navigation network (CCVNet) for SAR ship rotation detection, which uses an anchor-free method to directly predict the vector from the center to the corner, which can reduce the number of separate predictions' accumulation of errors caused by angles and scales. Cheng et al. [56] proposed an arbitrary orientation-oriented SAR ship detection network, global context-guided feature balance network (GFB-Net). The method obtains the output feature map by rotating ROIAlign, and uses it for classification and regression in the second stage of the network, and finally realizes ship detection in any direction. Zhao et al. [57] designed a head network with stepwise regression from coarse-grained to fine-grained to accurately detect ships in arbitrary orientations. At the same time, the method also combines the attention module to calibrate the multi-scale fusion features to highlight the ship information while suppressing the surrounding background interference. Li et al. [58] proposed a new directional SAR ship detector and embedded a hybrid convolutional channel attention (MCCA) module in the backbone network to enhance the ship by reweighting all channels of the feature map and boat characterization features. Zhao et al. [59] improved the original deep subdomain adaptation network (DSAN) and designed a dual branch network (DBN) embedded with an attention module to extract more discriminative deep transferable features, thereby improving the subdomain adaptation performance. Sun et al. [60] proposed an arbitrary orientation SAR ship detector (BiFA-YOLO) based on bidirectional feature fusion and angle classification. In order to effectively detect arbitrary orientation and densely distributed ships in HR SAR images, BiFA-YOLO adds an angle classification structure to the head network. He et al. [61] proposed a directional ship detector based on dual-branch probes and adaptive SAR feature enhancement, and an oriented region proposal network (ORPN) was designed to study the conversion of horizontal regions of interest to rotational regions of interest. Shao et al. [62] proposed a rotation-balanced feature alignment network (RBFA-Net) and designed an anchor-guided feature alignment network (AFAN) to adaptively align convolutional features according to the rotated anchor boxes. Xu et al. [63] proposed an arbitrary orientation-oriented SAR ship detection model combining triangle distance IoU loss (TDIoU loss) and attention-weighted feature pyramid network (AW-FPN), and proposed TDIoU loss as a rotating bounding box regression efficient solution for inconsistencies in loss metrics and discontinuities in boundaries.

Based on the above survey, it is not difficult to find that the current work of ship detection on SAR images mainly focuses on how to improve the performance of target detection in complex scenes while reducing the complexity of the model and improving the efficiency of the algorithm. This is also the starting point of our proposed method in this paper. However, different from the above research, we focus more on the comprehensive detection and recognition performance of ship targets.

## 3. Proposed Method

In this part, a thorough description of the suggested lightweight model will be given. First, the general structure of our model is shown. After that, each important module's mechanism will be described.

### 3.1. Overall Framework

Figure 1 depicts the process and general structure of the strategy suggested in this article. We choose the YOLOv5n model as the basic framework, which has a more concise network structure and faster running and reasoning speed, so it is easier to achieve lightweight deployment. Our improved method is mainly composed of three parts: the first part is the improved backbone network in the dotted frame in Figure 1. First of all, in order to effectively suppress the background noise and enhance the prominent target features, we embed a lightweight attention module in each C3 feature extraction module on the basis of the original YOLOv5-n backbone network framework. Second, based on the improvement of the original framework, the attention structure and the simplified spatial pyramid pooling fast structure (SimSPPF) were combined to further improve the generation speed of the target candidate box and effectively save the computational cost. In the part of prediction, we added the angle classification module to the prediction layer, and improved the constraint through the loss function to achieve the directed prediction of ship targets.

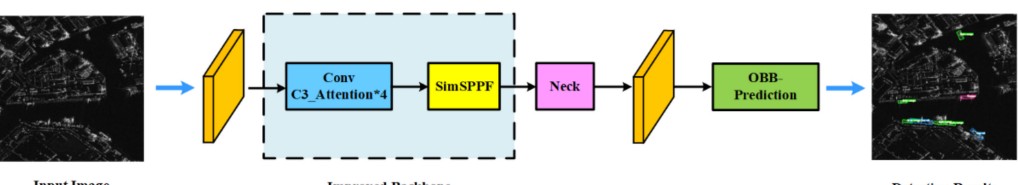

**Figure 1.** The overall framework of the proposed lightweight model.

### 3.2. C3_Attention Block

In contrast with the optical image, the ship target contains less information in the SAR image, while the background occupies most of the pixels. Therefore, in complex scenes such as inshore areas and high sea conditions, the ship features extracted by the network are less dominant. For this reason, we choose to add an attention module in the network feature extraction layer to further enhance the highlighting of target features and suppress the interference of background noise. Among many attention modules, we choose to embed the strong attention module squeeze excitation (SE) attention model and the convolutional block attention module (CBAM) based on the original C3 structure and construct new feature extraction structures C3SE and C3CBAM. Figure 2a,b show the structures of the C3SE and C3CBAM. Unlike transformer, which consumes a lot of computing resources to obtain the relationship between features, SE and CBAM modules directly calculate the feature weights in channel dimension and spatial dimension, and have significant advantages in computational efficiency and reasoning speed.

As seen in Figure 2a, C3SE initially decreases the dimension of spatial features by global average pooling based on the width and height of the feature map in order to draw emphasis to the channel dimension. Then, as seen in Equation (1), the link between the channels is established utilizing two complete connection layers and a nonlinear activation function.

$$F_c = T(ReLU(T(\frac{1}{H \times W}\sum_{i=1}^{H}\sum_{j=1}^{W}F_{in}(i,j)))) \tag{1}$$

where $F_{in}$ represents input features, $F_c$ represents middle layer features, $H$ and $W$ represent the width and height of the channel, respectively, $ReLU$ represents the nonlinear activation function, and $T$ represents the fully connected layer.

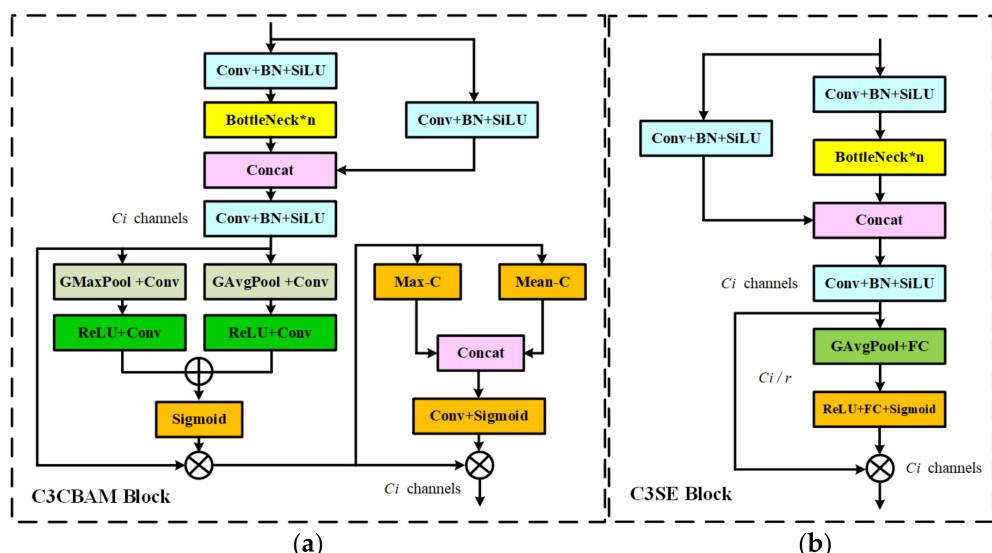

**Figure 2.** The structures of two proposed different attention blocks. (**a**) The C3SE block; (**b**) The C3CBAM block. "*Ci*" represents the number of feature channels, "Conv" means the ordinary convolution-2D layer, "BN" means the batch normalization layer, "SiLU" means swish function, "BottleNeck" is formed by stacking multiple convolution layers, "GAvgPool" means the global average pooling-2D layer; "GMaxPool" means the global max pooling-2D layer; "FC" means the fully connection layer, and the "r" is a reduction factor, "Max-C" represents the maximum-channel-pixel, "Mean-C" means the average-channel-pixel.

Following the Sigmoid activation function, the normalized weight is acquired, and it is then multiplied by each channel of the original feature graph to complete the channel attention and recalibrate the original feature, as given in Equation (2).

$$F_{out} = F_{in} \cdot \sigma(F_c) \tag{2}$$

where $F_{out}$ represents the output features and $\sigma$ represents the Sigmoid function. The global receptive field may be produced by global average pooling. By lowering the dimension of the feature graph, the parameters and computation in the first complete connection are significantly lowered. It is restored to its original number of channels by a complete connection after a nonlinear activation function, and the correlation between channels is formed. After that, the network can pay more attention to the channel characteristics that contain more information, and reduce the attention to the features with less information.

The C3CBAM attention mechanism module is separated into two sections: spatial attention and channel attention, as seen in Figure 2b. The featured graph is input, followed by the channel attention, GAP, and GMP based on the width and height of the feature graph, the attention weight for the channel through *MLP*, the normalized attention weight through the Sigmoid function, and finally the weighting of the normalized attention weight to the original input feature graph channel by the channel through multiplication. Complete Equation (3)'s formula for the original feature's channel attention recalibration:

$$F_c = \sigma(MLP(AvgPool(F_{in})) + MLP(MaxPool(F_{in}))) \tag{3}$$

where *AvgPool* represents the average pooling, *MaxPool* represents the maximum pooling, *MLP* represents a two-layer neural network.

The feature graph output through the channel also performs global maximum pooling and global average pooling based on the width and height of the feature graph, transforming the feature dimension from H × W to 1 × 1, then reducing the dimension of the feature graph after convolution kernel 7 × 7 and Relu activation function, and finally lifting it to the original dimension after a convolution. This process obtains the attention feature in

spatial dimension. Finally, the feature map standardized by Sigmoid activation function and the feature map of channel attention outputs merged to complete the recalibration of the feature graph in the space and channel dimensions, as shown in Equation (4).

$$F_{out} = \sigma(f^{7\times7}([AvgPool(F_c); MaxPool(F_c)])) \tag{4}$$

where $f^{7\times7}$ represents the convolution operation with $7 \times 7$ convolution kernel. In the spatial attention module, the global average pooling and maximum pooling obtain spatial attention features, and the correlation between spatial features is established through two convolutions, while keeping the input and output dimensions unchanged. Through the convolution operation with $7 \times 7$ convolution kernel, the parameters and the amount of calculation are greatly reduced, which is beneficial to the establishment of high-dimensional spatial feature correlation. The association between each feature in the channel and space is substantially improved after CBAM, which makes it easier to extract the target's useful characteristics. The new feature graph will also obtain attention weights in the channel and spatial dimensions.

### 3.3. SimSPPF Block

Spatial pyramid pooling (SPP) was originally proposed by He et al. [64] to solve the problem that the size of the input image cannot always meet the requirements of the input, which leads to the distortion of the network input image. When applied to the target detection network, SPP can effectively solve the problem of graph correlation repetitive feature extraction by convolution neural network, which greatly improves the speed of generating candidate boxes and saves computational costs. Specifically, as shown in Figure 3a, SPP first carries out convolution and normalization activation processing on the input feature, and then carries on the maximum value pool processing to the activation feature on three scales, and then splices the three scale features together with convolution normalization activation to obtain the features, as shown in Equation (5).

$$F_{out} = CBL[MaxPool^{5\times5}CBL(F_{in}); MaxPool^{9\times9}CBL(F_{in}); MaxPool^{13\times13}CBL(F_{in}); CBL(F_{in})] \tag{5}$$

where *CBL* represents the combination of "Conv", "BN", and "ReLU".

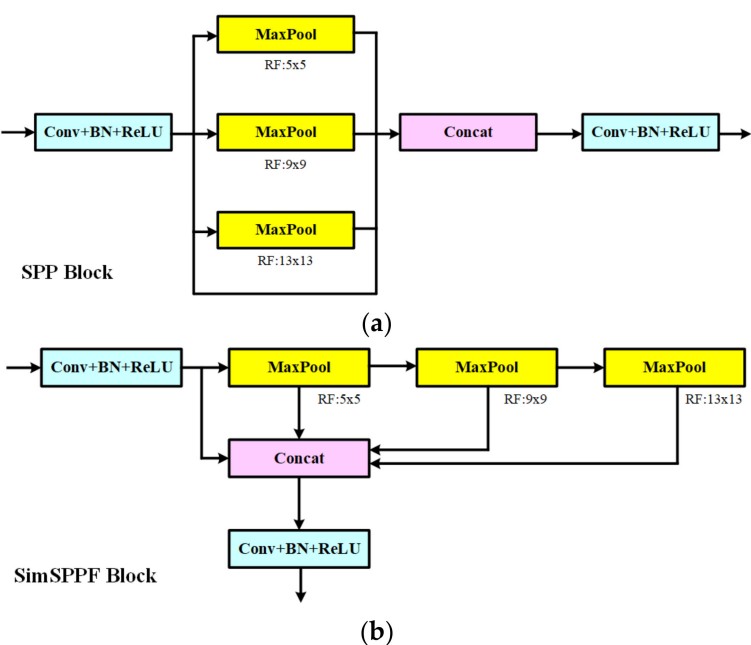

**Figure 3.** The structures of the SPP and SimSPPF block. "RF" represents the receptive field.

The SimSPPF module used in this paper is further improved and optimized based on the SPP structure. As shown in Figure 3b, SimSPPF also carries out the convolution and normalized activation of the input features at first, but the difference is that the activated features are not pooled at the same time at three scales, but progressive pooling. In this way, the efficiency of target feature extraction and candidate box selection can be further accelerated. The experimental results show that the processing speed of SimSPPF is significantly higher than that of SPP.

### 3.4. OBB Prediction Block

The original YOLOv5 network is based on rectangle regression. However, for the SAR images of complex scenes, there are many dense distributions of ship targets, especially in the nearshore scene, often docking many ships. As such, the network will suppress some high-quality prediction boxes when calculating the IOU between the target prediction frames. Therefore, the SAR ship detection method based on the original YOLOv5 network is not effective for ships close to the shore. In order to improve the detection performance, the regression mode of the network detection frame can be changed to the directed frame form. Due to the limitation of the angle, Sun et al. [60] classified and predicted ships in any direction by adding angle components to the structure of the network prediction layer. The experimental results show that the angle classification method can effectively improve the performance of ship target prediction without significantly increasing the complexity of the model. Figure 4 shows the improved rotation prediction head structure adopted in this paper.

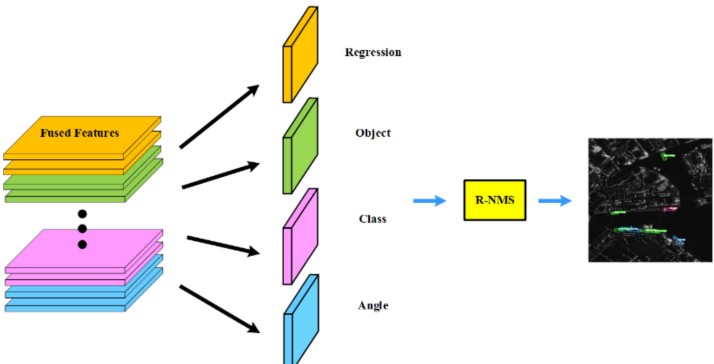

**Figure 4.** The structure of the oriented bounding box (OBB) prediction block.

Due to the addition of an angle classification module, the training loss function of the model proposed in this paper consists of four parts: classes loss, objectness loss, location loss, and angular loss [60], as shown in Equation (6):

$$Loss = L_{cls} + L_{obj} + L_{loc} + L_{ang} \tag{6}$$

where $L_{cls}$, $L_{obj}$, $L_{loc}$, and $L_{ang}$ denote the classes loss, objectness loss, location loss, and angular loss, respectively. Among them, the BCE loss is adopted for $L_{cls}$, $L_{obj}$, and $L_{ang}$, and the CIOU loss is adopted for $L_{loc}$. $L_{cls}$ represents the classification loss of the positive sample [16]. $L_{loc}$ represents the location loss of the positive sample. $L_{obj}$ represents the BCE loss between the prediction bounding box with the ground truth box. In order to balance the losses of different scales, the $L_{obj}$ of three prediction layers are dynamically weighted as Equation (7).

$$L_{obj} = \lambda_1 L_{obj}^{small} + \lambda_2 L_{obj}^{medium} + \lambda_3 L_{obj}^{large} \tag{7}$$

where $L_{obj}^{small}$, $L_{obj}^{medium}$, and $L_{obj}^{large}$, denote the objectness loss of three scales in the prediction layers, $\lambda_1$, $\lambda_2$, and $\lambda_3$ are the equilibrium coefficient.

## 4. Experiments

We conducted several tests on the recently published ship detection and identification dataset, SRSDD-v1.0, to assess the efficacy of the approaches suggested in this work. The experimental dataset, setup, and assessment indicators will initially be covered in this section. We conducted ablation tests to confirm the efficiency of the suggested module on this foundation. Finally, the outcomes of our visualization experiment using SRSDD datasets are presented and statistically evaluated against those of alternative deep-learning techniques. The results of the experiments demonstrate that the suggested technique has the best detection performance and the shortest model size.

### 4.1. Dataset

The SRSDD-v1.0 dataset proposed by Lei et al. [28] is built using 30 large-scene images of the Chinese GF-3 spotlight (SL) model, all with an image resolution of 1 m. These large-scene images are cropped into 1024 × 1024-pixel patches. After cropping, the dataset contains 666 SAR image slices and a total of 2884 ships. It is worth noted that this dataset is the first publicly released dataset containing different types of SAR ships. Table 1 shows the information on several datasets. In all published datasets, SRSDD is the only SAR ship dataset with different resolutions and different categories. Therefore, we conduct ship detection and recognition experiments based on this dataset. Figure 5 shows the quantity distribution of six categories of ships in the SRSDD-v1.0 dataset. Figures 6 and 7, respectively, show the distribution of the length, width, and direction angle of the ships in the training and testing datasets. The overall distribution of the training and testing datasets is consistent. Figure 8 shows some examples of ships in the SRSDD data, including the six categories of Ore-oil, Bulk cargo, Finishing, LawEnforce, Dredger, and Container.

**Table 1.** Comparison of different SAR ship datasets.

| Dataset | Size (Pixel) | Image (Num) | Ship (Num) | Annotations | Resolution (m) | Categories |
|---------|-------------|-------------|------------|-------------|----------------|------------|
| SSDD | 190–668 | 1160 | 2586 | HBB | 1–15 | 1 |
| SSDD+ | 190–668 | 1160 | 2586 | OBB | 1–15 | 1 |
| Official-SSDD | 190–668 | 1160 | 2586 | Polygon | 1–15 | 1 |
| SAR-Ship-Dataset | 256 × 256 | 43,819 | 59,535 | HBB | 3–25 | 1 |
| Air-SARship-1.0 | 3000 × 3000 | 31 | 461 | HBB | 1, 3 | 1 |
| Air-SARship-2.0 | 1000 × 1000 | 300 | 2040 | HBB | 1, 3 | 1 |
| HRSID | 800 × 800 | 5604 | 16,951 | Polygon | 0.5, 1, 3 | 1 |
| LS-SSDD-v1.0 | 24,000 × 16,000 | 15 | 6015 | HBB | 5 × 20 | 1 |
| RSDD-SAR | 512 × 512 | 7000 | 10,263 | OBB | 2–20 | 1 |
| SRSDD-v1.0 | 1024 × 1024 | 666 | 2884 | OBB | 1 | **6** |

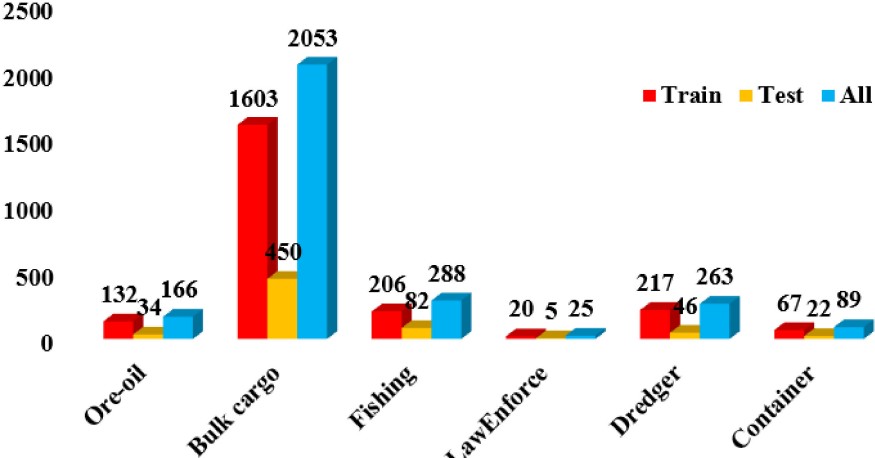

**Figure 5.** The quantity distributions of six categories of ships in SRSDD-v1.0 dataset.

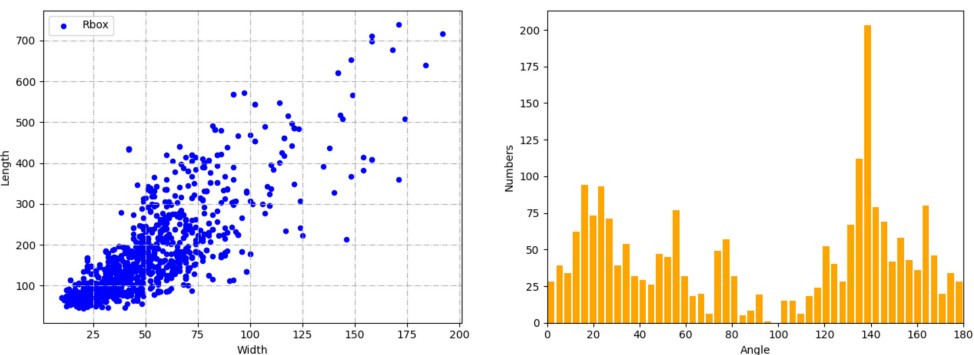

**Figure 6.** The distributions of the ships' width, length, and angle on the training dataset.

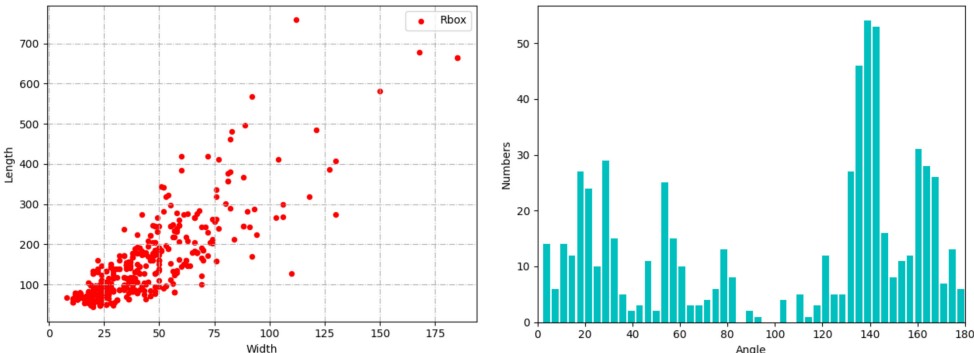

**Figure 7.** The distributions of the ships' width, length, and angle on the test dataset.

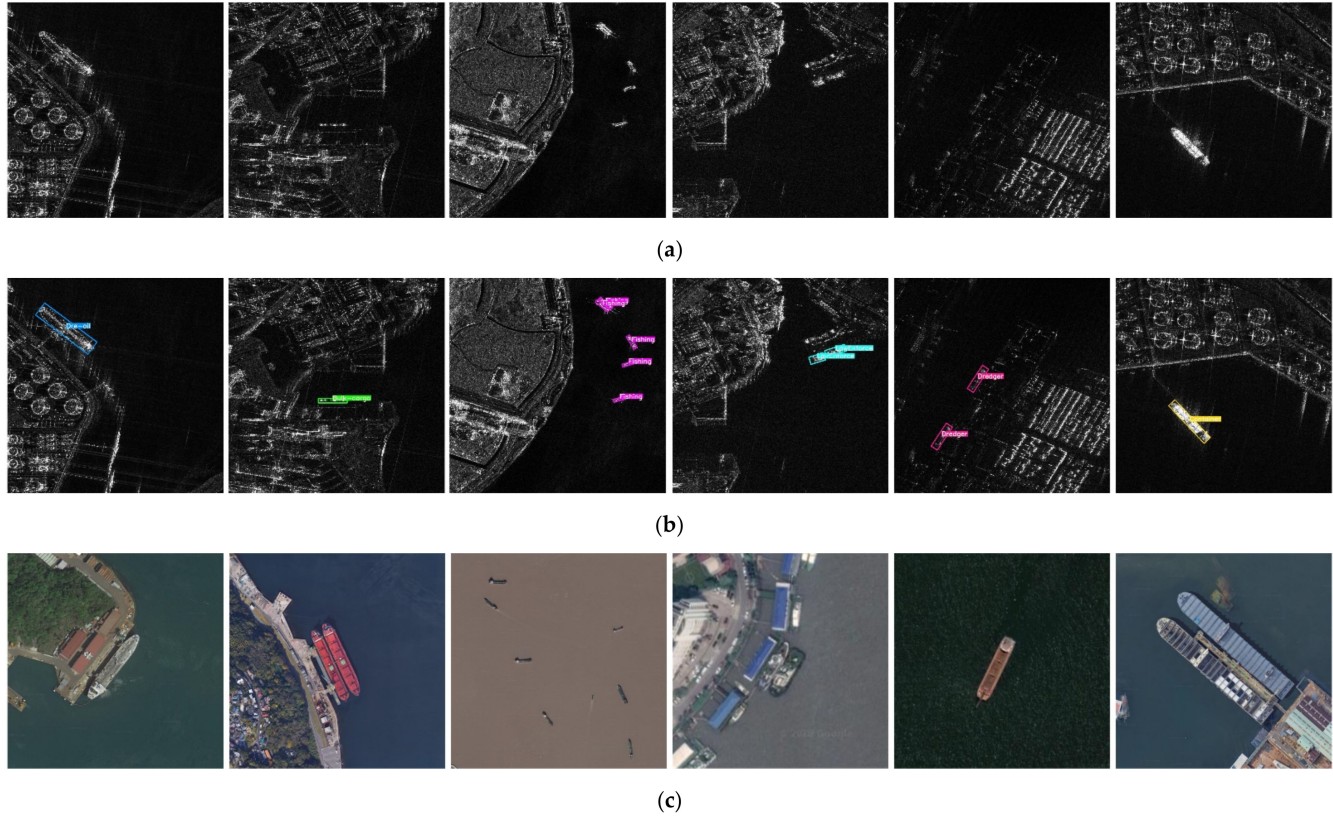

**Figure 8.** Instances of SRSDD-v1.0 dataset. (**a**) The original SAR images. (**b**) The ground truth. (**c**) The corresponding optical images. From left to right are the Ore-oil, Bulk cargo, Finishing, LawEnforce, Dredger, and Container.

### 4.2. Experimental Setup

All the tests in this work were carried out in PyTorch on a computer with an Intel Core (TM) i7-10875H CPU clocked at 2.30 GHz and an NVIDIA RTX 2070 GPU, as shown in Table 2. The computer was running in Windows 10. Table 2 shows how the computer and deep learning environment were set up for our experiments. In order to solve the problem of insufficient performance caused by the small sample size of the dataset, we used some traditional methods such as rotation and flipping to expand the original dataset to a certain extent. In addition, through the inspection of the data, we found that there were some problems with the original release data label, so we revised the data's label before training.

**Table 2.** Experimental setup and environment.

| Project | Model/Parameter |
| --- | --- |
| System | windows 10 |
| RAM | 32 GB |
| CPU | Intel i7-10875H |
| GPU | NVIDIA RTX 2070 |
| Platform | PyTorch |
| Code | python3.8 |
| Framework | CUDA10.1/cudnn7.6.5 |
| Epochs | 200 |
| Learning rate | 0.01 |
| Momentum | 0.0005 |

The stochastic gradient descent approach is used by the optimizer, and 200 training epochs are used. The learning rate is set to 0.01 and the momentum and weight decay rates are 0.0005 and 0.8, respectively. In every trial, the detection threshold IOU is set to 0.5.

### 4.3. Evaluation Metrics

We primarily use the accuracy, recall, and F1-score as assessment markers to compare various approaches in order to statistically measure the detection performance. The following are the definitions of recall and precision:

$$\text{Precision} = \frac{TP}{TP + FP} \tag{8}$$

$$\text{Recall} = \frac{TP}{TP + FN} \tag{9}$$

where the terms "true positives", "false positives", and "false negatives" stand for "true positives", "false positives", and "missing ships", respectively. Precision and recall are combined in the F1 score as follows:

$$F1 - score = 2 \times \frac{\text{Precision} \times \text{Recall}}{\text{Precision} + \text{Recall}} \tag{10}$$

Furthermore, we applied the frames-per second (FPS) to evaluate the detection efficiency of different methods as follows:

$$FPS = \frac{1}{T_{per-img}} \tag{11}$$

where $T_{per-img}$ represents the inference time per image. As for the algorithm complexity, we use the number of parameters, model size, and floating-point operations per second (FLOPS) to evaluate the different methods.

*4.4. Ablation Studies*

We performed a series of ablation tests on the SRSDD-v1.0 to evaluate the efficacy of each improvement suggested in this study.

### 4.4.1. Effect of C3_Attention Block

The suggested C3_Attention block module's effects on the effectiveness of model detection are examined through the tests in this section. Table 3 presents the experimental findings based on the test dataset. It appears that the suggested C3 Attention module can significantly enhance the detection performance, particularly the recall index. Finally, the C3SE module and the C3CBAM increase precision and recall by 2.69%, 2.88%, 3.77%, and 4.12%, respectively, in comparison to the baseline. Additionally, the F1 scores are higher by 3.24 and 3.51%, indicating that the suggested improved model may perform better at comprehensive identification. In terms of model complexity, the two proposed attention modules are lightweight models, so there is no increase in the final model size and FLOPS value.

**Table 3.** Performance comparison of different attention modules.

| Methods | Precision (%) | Recall (%) | F1 | FPS | Model (MB) | FLOPS |
|---------|---------------|------------|-----|-----|------------|-------|
| YOLOv5n (Base) | 55.42 | 53.77 | 54.58 | 75.19 | 4.06 | 4.2G |
| Base + C3SE | 58.11 | 57.54 | 57.82 | 73.00 | 4.06 | 4.2G |
| Base + C3CBAM | **58.30** | **57.89** | **58.09** | 72.46 | 4.06 | 4.2G |

### 4.4.2. Effect of SimSPPF Block

Table 4 shows the comparison of experimental results with or without the addition of SimSPPF modules. We can find that replacing SPP with SimSPPF in the backbone network structure will not affect the detection performance of the model, and the comprehensive F1 value of the model detection is still 54.48. In addition, the model complexity, including the total number of parameters, model size and computational complexity, remain unchanged. In terms of the operational efficiency of the model, the addition to SimSPPF structure, can significantly improve the reasoning speed, and FPS increases by 10.28 compared with the baseline.

**Table 4.** Performance comparison of with and without the SimSPPF Block.

| Methods | F1 | FPS | Param (M) | Model (MB) | FLOPS |
|---------|-----|-----|-----------|------------|-------|
| YOLOv5n (Base) | 54.58. | 75.19 | 1.68 | 4.06 | 4.2G |
| Base + SimSPPF | 54.58 | **85.47** | 1.68 | 4.06 | 4.2G |

### 4.4.3. Effect of OBB Prediction Block

In this section, the OBB prediction block is verified. The same parameter settings are used for training the network in these experiments. The results in Table 5 show that the OBB prediction method yields better overall performance. Specifically, prediction with OBB finally increases the precision and recall by 0.05% and 3.64%, respectively. Furthermore, the F1 scores are increased by 1.84, implying that the effectiveness of angle classification. To demonstrate the advantages of OBB prediction more visually, we visualized the HBB and OBB prediction results, as shown in Figure 9. Among them, different color callout boxes represent different categories. We can find that there are a large number of densely arranged ships in the SRSDD dataset, and there are obvious missing ships in the detection results of HBB prediction. On the contrary, the OBB prediction results are better able to capture a large number of docked or side-by-side ships.

**Table 5.** Performance comparison of different attention modules with OBB prediction method.

| Methods | Precision (%) | Recall (%) | F1 | FPS | Model (MB) | FLOPS |
|---|---|---|---|---|---|---|
| YOLOv5n (Base) | 55.42 | 53.77 | 54.58 | 75.19 | 4.06 | 4.2G |
| Base + OBB | 55.47 | **57.41** | **56.42** | 69.40 | 4.52 | 5.0G |

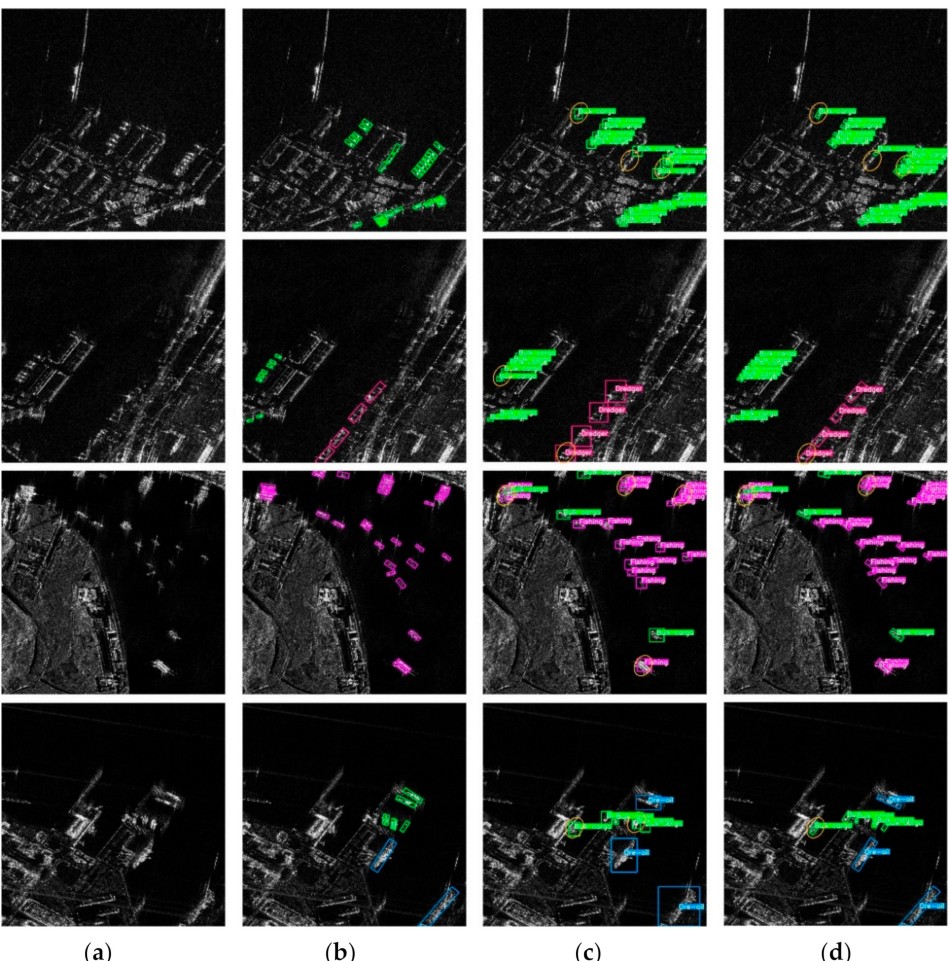

(**a**)　　　　　(**b**)　　　　　(**c**)　　　　　(**d**)

**Figure 9.** Detection results of different prediction methods on SRSDD. (**a**) The original SAR images. (**b**) The ground truth. (**c**) HBB prediction results. (**d**) OBB prediction results. Yellow ellipse indicates a false prediction.

### 4.5. Comparison with Other Methods

To verify the comprehensive performance of our method, we compare the proposed model with other deep-learning methods in terms of detection performance, inference efficiency, and model complexity (obtained from the corresponding literature), as shown in Table 6. It can be seen that, in the two-stage detection network, O-RCNN [65] has the best comprehensive detection performance in the SRSDD dataset, with the highest detection precision of 64.01%, a detection recall rate of 57.61%, and a comprehensive F1 value of 60.64. In the single-stage detection model, the comprehensive performance of BBAVectors [66] is the best, but the recall rate of this method on SRSDD data is only 34.56%. However, the F1 score of the proposed models in this paper is 61.26, which is 0.86% and 20.36% higher than that of O-RCNN and BBAVectors, respectively. More importantly, our method processes the SRSDD dataset almost seven times faster than other methods, and the model size is only 4.52 M, which is significantly better than the 244 M of the original lightest model R-FCOS [67]. Although the precision index of the proposed method is 59.70%, which is lower than the 64.01% of O-RCNN and 60.56% of R-FCOS. However, the recall index of

our model is 62.90%, which is significantly higher than the 57.61% of the O-RCNN and all other methods. The results show that our model can more effectively detect ship targets in complex-scene SAR images. In terms of the running efficiency of different models, the maximum FPS of two-stage detection models for processing images on the SRSDD dataset is only 8.38. Among the other one-stage detection models, R-RetinaNet [68] has the fastest processing speed, but the FPS of R-RetinaNet model is only 10.53. In contrast, through Table 6, we can clearly find that the FPS value of our model proposed in this paper is 68.02, and the processing efficiency has a significant advantage over other one-stage or two-stages detection models.

**Table 6.** Detection results of different CNN-based methods on SRSDD-v1.0.

| Model | Category | Precision (%) | Recall (%) | F1 | FPS | Model (M) |
|---|---|---|---|---|---|---|
| FR-O [28] | Two-stage | 57.12 | 49.66 | 53.13 | 8.09 | 315 |
| ROI [28,69] | Two-stage | 59.31 | 51.22 | 54.97 | 7.75 | 421 |
| Gliding Vertex [28,70] | Two-stage | 57.75 | 53.95 | 55.79 | 7.58 | 315 |
| O-RCNN [28,65] | Two-stage | **64.01** | 57.61 | 60.64 | 8.38 | 315 |
| R-RetinaNet [28,68] | One-stage | 53.52 | 12.55 | 20.33 | 10.53 | 277 |
| R3Det [28,71] | One-stage | 58.06 | 15.41 | 24.36 | 7.69 | 468 |
| BBAVectors [28,66] | One-stage | 50.08 | 34.56 | 40.90 | 3.26 | 829 |
| R-FCOS [28,67] | One-stage | 60.56 | 18.42 | 28.25 | 10.15 | 244 |
| Ours | One-stage | 59.70 | **62.90** | **61.26** | **68.02** | **4.52** |

### 4.6. Detection and Recognition Results on SRSDD

Finally, we visualize the detection and recognition results of the proposed model in the nearshore and offshore scenarios of the SRSDD dataset, as shown in Figures 10 and 11. From Figure 10, we can see that the proposed method can basically detect the inshore ships and assign the correct labels. This is mainly because the embedded attention module can strengthen the target features while suppressing the background interference information. For the nearshore scene, the background interference is serious, and the use of the attention enhancement feature can effectively remove the false alarms. Through Figure 11, it can be found that almost all the offshore ship targets are detected by the proposed method, which further verifies the effectiveness of our designed model.

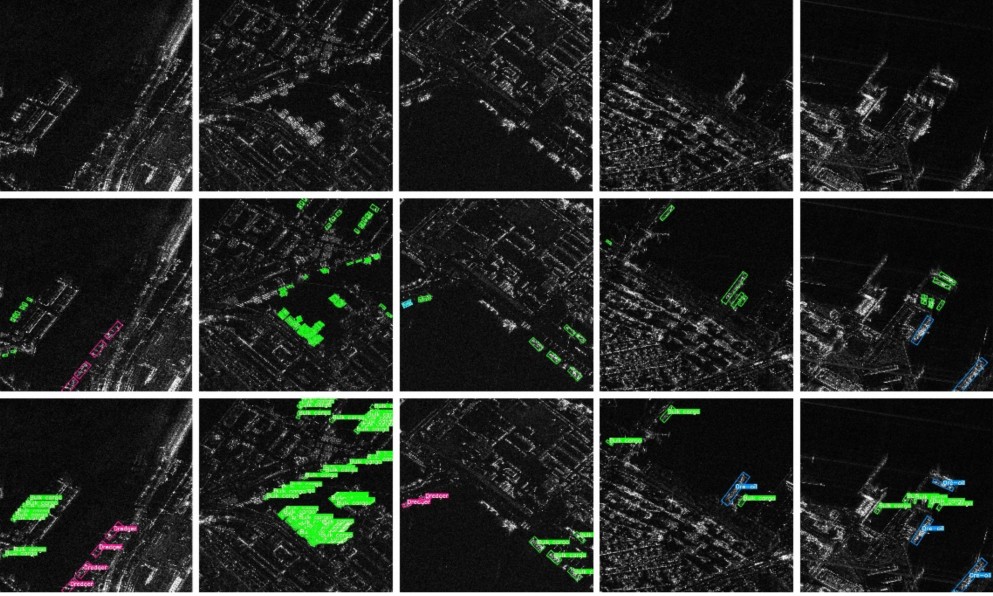

**Figure 10.** Detection and recognition results of inshore ships. The first column is the original SAR images, the second column is the ground truth, and the third column is our results.

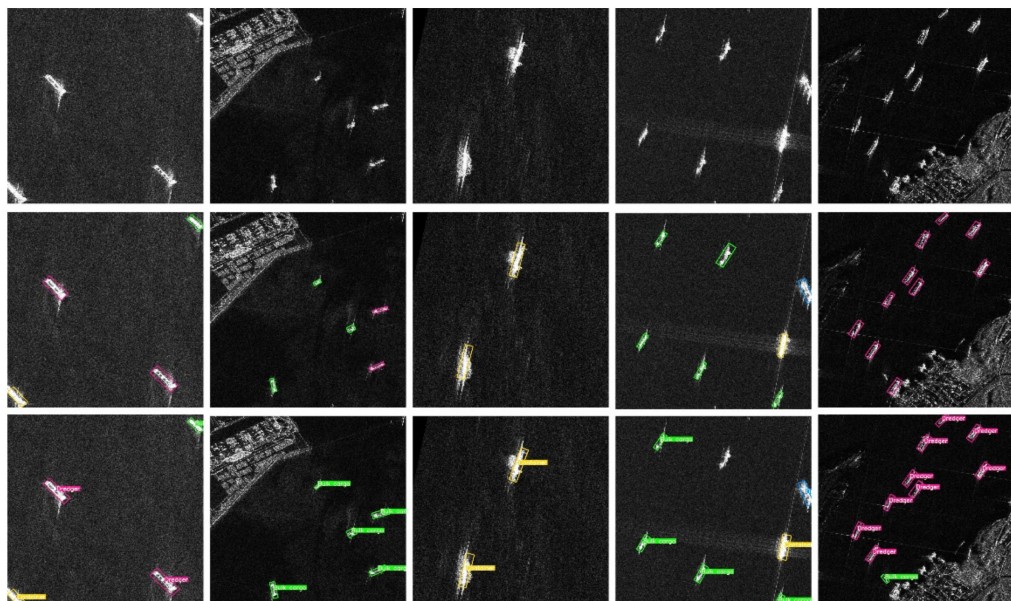

**Figure 11.** Detection and recognition results of offshore ships. The first column is the original SAR images, the second column is the ground truth, and the third column is our results.

In order to quantitatively analyze the detection and recognition performance of our method on SRSDD data, we calculate the detection and recognition results of six categories of ships, as shown in Tables 7 and 8. As can be seen from Table 7, the detection rates vary greatly among different categories, in which the comprehensive F1 value of Container can reach 0.71, but that of Fishing is only 0.39. This phenomenon demonstrates that our method still has obvious space for improvement in detection performance. In terms of target recognition performance, as shown in Table 8, the recognition rate of large ships such as Container, Dredger, and Ore-oil is significantly higher than that of small ships such as Fishing and Bulk cargo. Among them, the Background FN in the meter indicates that the false alarms in the detection and recognition results are non-ship targets, which mainly includes land interferers and sea islands. In other words, there is still some room for improvement in the detection performance on all categories of ships. In addition, the results of Table 8 show that the recognition performance of Bulk Cargo, Fishing and LawEnforce is relatively low. This is because Bulk Cargo accounts for the vast majority of the SRSDD dataset, while Fishing and LawEnforce are relatively few. Although this paper adopts some data augmentation methods to improve the recognition performance, the serious data imbalance results in the low recognition rate of these three categories of ships. On the other hand, Bulk Cargo, Fishing, and LawEnforce are mostly distributed along the shore, and the target size is small and often densely distributed, which brings some challenges to network detection and recognition. In any case, the method proposed in this paper can effectively detect and identify six categories of ships to a certain extent, and the current gap is also the focus of our future work.

**Table 7.** Detection results of six categories of ships.

| Class | Precision (%) | Recall (%) | F1 |
|---|---|---|---|
| Ore-oil | 53.5 | 46.7 | 0.50 |
| Bulk cargo | 52.6 | 59.3 | 0.56 |
| Fishing | 64.3 | 28.0 | 0.39 |
| LawEnforce | 44.2 | **100.0** | 0.61 |
| Dredger | **77.4** | 67.0 | **0.72** |
| Container | 66.4 | 76.2 | 0.71 |

**Table 8.** The confusion matrix of proposed method in the six categories of ships.

|  | Ore-Oil | Bulk Cargo | Fishing | LawEnforce | Dredger | Container | Background FN |
|---|---|---|---|---|---|---|---|
| Ore-oil | 0.47 | 0.01 | 0.00 | 0.00 | 0.02 | 0.00 | 0.02 |
| Bulk cargo | 0.00 | 0.53 | 0.13 | 0.00 | 0.10 | 0.10 | 0.78 |
| Fishing | 0.00 | 0.00 | 0.22 | 0.00 | 0.00 | 0.00 | 0.08 |
| LawEnforce | 0.00 | 0.00 | 0.00 | 0.25 | 0.00 | 0.00 | 0.04 |
| Dredger | 0.00 | 0.00 | 0.00 | 0.00 | 0.63 | 0.00 | 0.03 |
| Container | 0.00 | 0.00 | 0.00 | 0.00 | 0.00 | 0.71 | 0.04 |
| Background FN | 0.53 | 0.45 | 0.64 | 0.75 | 0.24 | 0.19 | 0.00 |

## 5. Conclusions

This research suggests a lightweight model for ship detection and recognition in complex-scene SAR images. In order to enhance the model's detection and recognition performance in complicated scenarios and increase the model's deployment possibilities, we optimized the module based on three aspects: the designed C3_attention module, the improved SimSPPF module, and the OBB prediction module. The results of many ablation experiments performed on the latest SRSDD-v1.0 dataset demonstrate the potency of each module created in this paper. Experimental findings demonstrate that the model suggested in this paper can perform well, with an F1-Score of 61.26 and an FPS of 68.02 on SRSDD, and using just 1.92 M parameters and 4.52 MB of model memory. It is worth noting that our approach is more suited for practical equipment than previous approaches and can meet the real-time requirements of SAR ship detection and recognition in the future. However, through many comparative experiments, we find that our method still has obvious deficiencies in the performance of ship target recognition due to the influence of image quality and resolution. Therefore, we will consider carrying out more in-depth research on the following aspects: (1) We will build a higher-quality SAR ship detection and recognition dataset including more categories of ships; (2) We will conduct more research on the improvement of SAR image quality to improve the performance of ship target recognition; (3) We will continue to investigate the study on the quick detection and recognition of ship targets in SAR images.

**Author Contributions:** Conceptualization, B.X.; methodology, B.X.; validation, B.X. and Z.S.; formal analysis, B.X. and Z.S.; investigation, B.X. and Z.S.; resources, Z.S.; data curation, Z.S. and J.W.; writing—original draft preparation, B.X. and Z.S.; writing—review and editing, J.W. and X.L.; visualization, Z.S.; supervision, K.J.; project administration, K.J.; funding acquisition, X.L. All authors have read and agreed to the published version of the manuscript.

**Funding:** This work was jointly supported by National Natural Science Foundation of China (62001480) and the Hunan Provincial Natural Science Foundation of China (2021JJ40684).

**Institutional Review Board Statement:** Not applicable.

**Informed Consent Statement:** Not applicable.

**Data Availability Statement:** Not applicable.

**Acknowledgments:** The authors would like to thank the pioneer researchers in SAR ship detection and other related fields.

**Conflicts of Interest:** The authors declare no conflict of interest.

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
