# Peer review of "A Lightweight Model for Ship Detection and Recognition in Complex-Scene SAR Images"

_remotesensing, doi:10.3390/rs14236053_

Round 1

Reviewer 1 Report

find the attached file

Author Response

The authors would like to express our sincere appreciation to the reviewer for his/her time and effort spent in evaluating our work and his/her constructive comments that helped us to improve the quality and presentation of the manuscript further. We are also most appreciative of the reviewer’s encouraging feedback “The motivation is clear and the manuscript is well-structured. Experiments prove the effectiveness of the method well.”

We have highlighted all the changes in blue color in our revised manuscript. We hope that the revised manuscript addresses the reviewers’ concerns in a satisfactory way.

Reviewer 2 Report

The paper proposes a new architecture for ship detection using SAR images that is faster than existing architectures. The algorithm benefits from YOLOv5-n as the backbone and relies on a pyramid pooling for fast and efficient feature extraction. Experiments on the SRSDD dataset are provided to demonstrate that the method ie effective and leads to improved performance.   The paper reads well and the results are convincing. I have the following comments to be incorporated for the next revision:     1. Please add more explanations about the loss functions you use for training your architecture.   2. Could please provide a paragraph and explain the difference between your attention block and transformers?   3. Efficiency is an important claim in this paper, I would like to see the running time added to Table 6, so we can see that the proposed method is indeed faster.   4. There are several recent works that benefit from efficient deep learning to address ship detection:   a. Zhang, S., Wu, R., Xu, K., Wang, J. and Sun, W., 2019. R-CNN-based ship detection from high resolution remote sensing imagery. Remote Sensing11(6), p.631.   b. Chang, Y.L., Anagaw, A., Chang, L., Wang, Y.C., Hsiao, C.Y. and Lee, W.H., 2019. Ship detection based on YOLOv2 for SAR imagery. Remote Sensing11(7), p.786.   c. Rostami, M., Kolouri, S., Eaton, E. and Kim, K., 2019. Deep transfer learning for few-shot SAR image classification. Remote Sensing11(11), p.1374.   d. Zhang, Y., Guo, L., Wang, Z., Yu, Y., Liu, X. and Xu, F., 2020. Intelligent ship detection in remote sensing images based on multi-layer convolutional feature fusion. Remote Sensing12(20), p.3316.   e. Chen, L., Shi, W. and Deng, D., 2021. Improved YOLOv3 based on attention mechanism for fast and accurate ship detection in optical remote sensing images. Remote Sensing13(4), p.660.   These works should be discussed in the related work section for thorough coverage of progress in adopting deep learning in ship detection.   5. In Table 7, can you add other methods for comparison? I would like to see whether other methods have a behavior similar to the proposed architecture.   6. Please run your code several times, e.g., 10 times, and report both the average and the standard variation to make the comparison statistically meaningful.   7. Is it possible to release the code on a public domain like GitHub for the reproducibility of results by other researchers?

Author Response

The authors would like to express our sincere appreciation to the reviewer for his/her time and effort spent in evaluating our work and his/her constructive comments that helped us to improve the quality and presentation of the manuscript further. We are also most appreciative of the reviewer’s encouraging feedback “The paper reads well and the results are convincing.”

We have highlighted all the changes in blue color in our revised manuscript. We hope that the revised manuscript addresses the reviewers’ concerns in a satisfactory way.

Reviewer 3 Report

See the attached file for detail comments.

Author Response

The authors would like to express our sincere appreciation to the reviewer for his/her time and effort spent in evaluating our work and his/her constructive comments that helped us to improve the quality and presentation of the manuscript further. We are also most appreciative of the reviewer’s encouraging feedback “The contribution is obvious and the presentation is plausible.”

We have highlighted all the changes in blue color in our revised manuscript. We hope that the revised manuscript addresses the reviewers’ concerns in a satisfactory way.

Round 2

Reviewer 1 Report

overall the quality of the paper is better than the first version and the author has answered all of my questions, I have no more questions, and this paper can be published in its current form.

Reviewer 2 Report

The authors have addressed my concerns decently. My only feedback is to release the code by the publication of the paper. This will help the paper to be recognized.